# Radial-VCReg: More Informative Representation Learning Through Radial Gaussianization

**Yilun Kuang**[*]
New York University

**Yash Dagade**
Duke University

**Deep Chakraborty**
University of Massachusetts Amherst

**Erik Learned-Miller**
University of Massachusetts Amherst

**Randall Balestriero**
Brown University

**Tim G. J. Rudner**
University of Toronto

**Yann LeCun**[*]
New York University

## Abstract

Self-supervised learning aims to learn maximally informative representations, but explicit information maximization is hindered by the curse of dimensionality. Existing methods like VCReg address this by regularizing first- and second-order feature statistics, which cannot fully achieve maximum entropy. We propose Radial-VCReg, which augments VCReg with a radial Gaussianization loss that aligns feature norms with the Chi distribution—a defining property of high-dimensional Gaussians. We prove that Radial-VCReg transforms a broader class of distributions toward normality compared to VCReg and show on synthetic and real-world datasets that it consistently improves performance by reducing higher-order dependencies and promoting more diverse and informative representations.

## 1 Introduction

Self-supervised learning leverages unlabeled data to create useful representations for downstream tasks [Radford et al., 2018, Chen et al., 2020]. Many methods are based on the InfoMax principle, which aims to maximize mutual information between different views of the same input [Hjelm et al., 2019, Ozsoy et al., 2022]. This requires both enforcing agreement across views and preserving feature diversity to prevent collapse—the latter being more challenging.

Non-contrastive self-supervised learning methods like the VCReg component of VICReg [Bardes et al., 2022] address this by regularizing the covariance of features [Zbontar et al., 2021, Ermolov et al., 2021, Bardes et al., 2022]. While effective in practice [Sobal et al., 2025], covariance regularization only removes linear dependencies and cannot fully maximize information.

In this paper, we aim to optimize the InfoMax objective by *Gaussianizing* feature representations. The Gaussian distribution is the maximum entropy distribution for a given mean and variance [Cover and Thomas, 1991], encouraging features to be maximally spread out and resistant to collapse. Unfortunately, directly matching the feature distribution to a high-dimensional Gaussian suffers from the curse of dimensionality. Previous methods such as E2MC circumvent this by maximizing entropy per feature dimension along with whitening [Chakraborty et al., 2025]. However, there exist distributions that minimize the E2MC loss but do not maximize entropy. See Figure 1a for example.

In this work, we propose to Gaussianize our features radially. A $d$-dimensional isotropic Gaussian concentrates on a thin shell of radius $\sqrt{d}$ with an $O(1)$ width, whose marginal follows a Chi

---

[*]Corresponding authors. Email: `yilun.kuang@nyu.edu`, `yann.lecun@nyu.edu`.

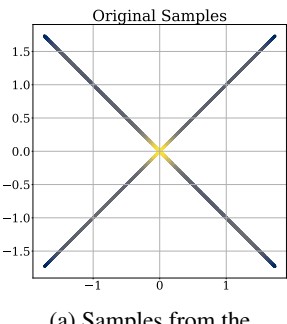 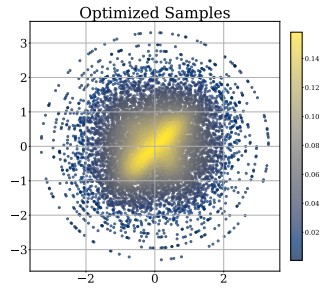 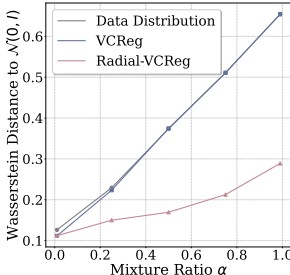

| (a) Samples from the X-Distribution. | (b) Samples optimized with the Radial-VCReg objective. | (c) Wasserstein Distance between optimized samples and $\mathcal{N}(\mathbf{0}, \mathbf{I})$. |

Figure 1: **The Radial-VCReg objective more effectively pushes samples from a non-elliptically symmetric X-distribution towards the standard normal distribution in 2D compared to the VCReg objective.** (a) The X-distribution has an identity covariance matrix, but it is not elliptically symmetric. (b) Samples from the X-distribution are optimized with the Radial-VCReg loss, yielding a spherical structure. (c) As the ratio $\alpha$ of samples from the X-distribution increases, samples optimized with the Radial-VCReg loss are closer to the standard normal compared to that of VCReg. The VCReg objective is also unable to move the samples away from their starting distributions.

distribution [Vershynin, 2018]. Enforcing this radial property with whitening provides sufficient conditions for Gaussianity if the underlying distribution is elliptically symmetric [Lyu and Simoncelli, 2009, 2008].

Inspired by this observation, we explore to what extent we can obtain Gaussian features in self-supervised learning by enforcing a chi-distribution on the radial marginal of neural network features to maximize information. To summarize, our main contributions are as follows:

1. We propose the Radial Gaussianization loss, a consistent estimator of the Kullback–Leibler divergence between the empirical radius distribution and the ground-truth chi-distribution.

2. We introduce Radial-VCReg, a self-supervised method that extends VCReg by explicitly regularizing radial distributions, with theoretical guarantees of transforming a broader class of feature distributions toward normality.

3. We demonstrate empirically that Radial-VCReg 1) pushes the sample distributions closer to the standard normal compared to VCReg in synthetic settings, even in cases where the underlying distribution might not be elliptically symmetric, and 2) achieves consistent gains on real-world image datasets over VCReg.

Our codes are available at https://github.com/YilunKuang/RadialVCReg

## 2 Radial Gaussianization

In the following section, we show how to incorporate radial Gaussianization into an optimization objective for self-supervised learning. Additional background can be found in Appendix 6.

### 2.1 Self-Supervised Learning

In self-supervised learning, we are given unlabeled samples $\mathbf{X} = [\mathbf{x}_1, \cdots, \mathbf{x}_N]$ drawn from a data distribution $p_X$, where $\mathbf{x}_i \in \mathbb{R}^{d_{\text{in}}}$ and $\mathbf{X} \in \mathbb{R}^{N \times d_{\text{in}}}$. During training, we sample transformations $t, t' \sim \mathcal{T}$ and apply them to the original samples to create two sets of transformed samples, $\mathbf{X}_{\text{aug}} = [t(\mathbf{x}_1), \cdots, t(\mathbf{x}_N)]$ and $\mathbf{X}'_{\text{aug}} = [t'(\mathbf{x}_1), \cdots, t'(\mathbf{x}_N)]$, which form positive pairs. The goal is to train a neural network $h_{\boldsymbol{\theta}}$ to learn representations such that the resulting positive pairs, $\mathbf{Z} = [h_{\boldsymbol{\theta}}(t(\mathbf{x}_1)), \cdots, h_{\boldsymbol{\theta}}(t(\mathbf{x}_N))]$ and $\mathbf{Z}' = [h_{\boldsymbol{\theta}}(t'(\mathbf{x}_1)), \cdots, h_{\boldsymbol{\theta}}(t'(\mathbf{x}_N))]$, are close according to a specified distance metric. Simultaneously, the output features $\mathbf{z}_i, \mathbf{z}'_i \in \mathbb{R}^{d_{\text{out}}}$ must remain diverse and informative, avoiding representational collapse.

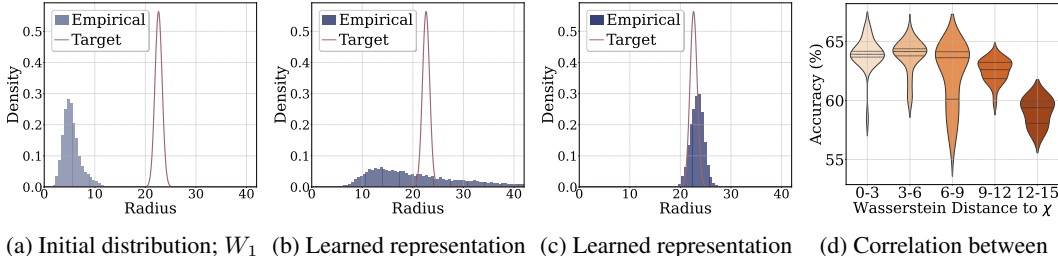

(a) Initial distribution; $W_1$ dist to $\chi = 17.15$    (b) Learned representation with VICReg; $W_1$ dist to $\chi = 8.17$    (c) Learned representation with Radial-VICReg; $W_1$ dist to $\chi = 0.79$    (d) Correlation between $\chi$-match and accuracy.

Figure 2: **Radial-VICReg enforces a chi-distributed radius after optimization, and there exists a correlation between classification accuracy and the quality of the chi-distribution matching.** (a) The feature norm distribution at random initialization with Wasserstein distance $W_1$ to the Chi distribution $\chi$ equal to 17.15. (b) Feature norm distribution under the VICReg loss is far away from the Chi distribution. (c) Representations learned with Radial-VICReg is closely matching the Chi distribution density function. (d) Across hyperparameter sweeps, validation accuracy increases as the radii distribution better matches the $\chi$-distribution as measured by lower Wasserstein distance.

## 2.2 Variance-Invariance-Covariance Regularization (VICReg)

VICReg [Bardes et al., 2022] is a non-contrastive self-supervised learning method that contains the variance, invariance, and covariance loss terms. For a feature matrix $\mathbf{Z} \in \mathbb{R}^{N \times d_{\text{out}}}$, we denote the $i$-th row as $\mathbf{z}_i \in \mathbb{R}^{d_{\text{out}}}$ and the $j$-th column as $\mathbf{z}^j \in \mathbb{R}^N$. The variance loss is given by $v(\mathbf{Z}) = \frac{1}{d_{\text{out}}} \sum_{j=1}^{d_{\text{out}}} \max(0, \gamma - \sqrt{\text{Var}(\mathbf{z}^j) + \epsilon})$, where $\gamma$ is typically fixed at 1. The invariance loss, computed as the mean squared error between $\mathbf{Z}$ and $\mathbf{Z}'$, is given by $s(\mathbf{Z}, \mathbf{Z}') = \frac{1}{N} \sum_{i=1}^{N} \|\mathbf{z}_i - \mathbf{z}'_i\|_2^2$. This term encourages positive pairs to have similar representations. Let the empirical covariance matrix for the feature matrix $\mathbf{Z}$ be $C(\mathbf{Z}) = \frac{1}{N-1} \sum_{i=1}^{N} (\mathbf{z}_i - \bar{\mathbf{z}})(\mathbf{z}_i - \bar{\mathbf{z}})^\top$, where $\bar{\mathbf{z}} = \frac{1}{N} \sum_{i=1}^{N} \mathbf{z}_i$ is the empirical mean. The covariance loss is defined as $c(\mathbf{Z}) = \frac{1}{d_{\text{out}}} \sum_{i \neq j} [C(\mathbf{Z})]_{i,j}^2$. Combining the three terms, we arrive at the VICReg formulation:

$$\mathcal{L}_{\text{VICReg}}(\mathbf{Z}, \mathbf{Z}') = \lambda_1 s(\mathbf{Z}, \mathbf{Z}') + \lambda_2 [v(\mathbf{Z}) + v(\mathbf{Z}')] + \lambda_3 [c(\mathbf{Z}) + c(\mathbf{Z}')] \tag{1}$$

where the variance and covariance losses are applied to $\mathbf{Z}$ and $\mathbf{Z}'$ separately. When $\lambda_1 = 0$, we call it VCReg.

## 2.3 Radial-VICReg

Let $\|\mathbf{z}\|_2$ be the norm (or radius) of the feature vector $\mathbf{z}$ with density $p_{\boldsymbol{\theta}}(\|\mathbf{z}\|_2)$. Our goal is to minimize the Kullback–Leibler divergence between $p_{\boldsymbol{\theta}}(\|\mathbf{z}\|_2)$ and the Chi-distribution $p_\chi(\|\mathbf{z}\|_2)$:

$$\min_{\boldsymbol{\theta}} D_{KL}\left( p_{\boldsymbol{\theta}}(\|\mathbf{z}\|_2) \,\middle\|\, p_\chi(\|\mathbf{z}\|_2) \right) = \underbrace{\mathbb{E}_{\|\mathbf{z}\|_2 \sim p_{\boldsymbol{\theta}}(\|\mathbf{z}\|_2)}[-\log p_\chi(\|\mathbf{z}\|_2)]}_{\text{Cross-Entropy}} - \underbrace{H(p_{\boldsymbol{\theta}}(\|\mathbf{z}\|_2))}_{\text{Entropy}} \tag{2}$$

where the cross entropy term is approximated using the Monte Carlo estimate:

$$\mathbb{E}_{\|\mathbf{z}\|_2 \sim p_{\boldsymbol{\theta}}(\|\mathbf{z}\|_2)}[-\log p_\chi(\|\mathbf{z}\|_2)] = \mathbb{E}\left[ \underbrace{\left(\frac{d}{2} - 1\right)\log 2 + \log \Gamma\left(\frac{d}{2}\right)}_{\text{constants}} + \frac{\|\mathbf{z}\|_2^2}{2} - (d-1)\log\|\mathbf{z}\|_2 \right] \tag{3}$$

$$\approx \frac{\beta_1}{N} \sum_{i=1}^{N} \left( \frac{1}{2}\|\mathbf{z}_i\|_2^2 - (d_{\text{out}} - 1)\log\|\mathbf{z}_i\|_2 \right) + C \tag{4}$$

with a tunable hyperparameter $\beta_1$ and an irrelevant constant $C$. The entropy term can also be computed using the m-spacing estimator [Vasicek, 1976, Learned-Miller et al., 2003]:

$$H(p_{\boldsymbol{\theta}}(\|\mathbf{z}\|_2)) \approx \frac{\beta_2}{N-m} \sum_{i=1}^{N-m} \log\left( \frac{N+1}{m}\left( \|\mathbf{z}_{(i+m)}\|_2 - \|\mathbf{z}_{(i)}\|_2 \right) \right) \tag{5}$$

where $\beta_2$ is a tunable hyperparameter, $m$ is the spacing hyperparameter, and $\|\mathbf{z}_{(1)}\|_2 \leq \|\mathbf{z}_{(2)}\|_2 \leq \cdots \leq \|\mathbf{z}_{(N)}\|_2$ are the ordered samples of the set $\{\|\mathbf{z}_i\|_2\}_{i=1}^N$. We refer to the composition of the cross-entropy and entropy loss as the radial Gaussianization loss $r(\mathbf{Z}; \beta_1, \beta_2)$.

$$r(\mathbf{Z}; \beta_1, \beta_2) = \frac{\beta_1}{N} \sum_{i=1}^{N} \left( \tfrac{1}{2}\|\mathbf{z}_i\|_2^2 - (d_{\text{out}} - 1) \log \|\mathbf{z}_i\|_2 \right) \tag{6}$$

$$- \frac{\beta_2}{N-m} \sum_{i=1}^{N-m} \log\left( \tfrac{N+1}{m} (\|\mathbf{z}_{(i+m)}\|_2 - \|\mathbf{z}_{(i)}\|_2) \right) \tag{7}$$

By the Law of Large Numbers, the cross-entropy estimator is consistent. Vasicek [1976] also shows that the m-spacing estimator is consistent. If $\beta_1$ and $\beta_2$ are both set to 1, the radial Gaussianization loss is a consistent estimator of the true Kullback–Leibler divergence between $p_\theta(\|\mathbf{z}\|_2)$ and the Chi distribution $p_\chi(\|\mathbf{z}\|_2)$ up to constant offsets, as it is a linear combination of two consistent estimators. When $\beta_1$ and $\beta_2$ are not equal to 1, the loss no longer corresponds exactly to the KL divergence; instead, it yields a weighted variant of the objective, similar in spirit to the modification introduced in $\beta$-VAE [Higgins et al., 2017].

In practice, we apply this term to both $\mathbf{Z}$ and $\mathbf{Z}'$, resulting in the Radial-VICReg loss:

$$\mathcal{L}_{\text{Radial-VICReg}}(\mathbf{Z}, \mathbf{Z}') = \mathcal{L}_{\text{VICReg}}(\mathbf{Z}, \mathbf{Z}') + r(\mathbf{Z}; \beta_1, \beta_2) + r(\mathbf{Z}'; \beta_1, \beta_2) \tag{8}$$

We notice that sometimes it's useful to include a multiplicative term $1/d_{\text{out}}$ for the cross entropy term, but we view this as absorbed in the $\beta_1$ hyperparameter. The goal of radial Gaussianization can also be achieved with other optimization objectives. We defer the details on alternative loss constructions to Appendix 9.

In Proposition 1, we show that the set of distributions Gaussianizable by Radial-VCReg (with $\lambda_1 = 0$) strictly contains that of VCReg. (See Appendix 7 for proofs.) Thus, we interpret the radial Gaussianization term as enforcing a necessary—but not sufficient—condition for Gaussianity.

**Proposition 1.** *Let $\mathbf{X}$ be a random vector in $\mathbb{R}^d$ with distribution $P_{\mathbf{X}}$. Define the VCReg map and Radial-VCReg map as*

$$T_{\text{VCReg}}(\mathbf{x}) = \mathbf{\Sigma}^{-1/2}(\mathbf{x} - \boldsymbol{\mu}) \tag{9}$$

$$T_{\text{Radial-VCReg}}(\mathbf{x}) = \frac{\mathbf{\Sigma}^{-1/2}(\mathbf{x} - \boldsymbol{\mu})}{\|\mathbf{\Sigma}^{-1/2}(\mathbf{x} - \boldsymbol{\mu})\|_2} F_\chi^{-1}\left( F_{\|\mathbf{\Sigma}^{-1/2}(\mathbf{x}-\boldsymbol{\mu})\|_2}(\|\mathbf{\Sigma}^{-1/2}(\mathbf{x} - \boldsymbol{\mu})\|_2) \right) \tag{10}$$

*where $\boldsymbol{\mu} = \mathbb{E}[\mathbf{X}]$, $\mathbf{\Sigma} = \text{Cov}[\mathbf{X}]$, $F_{\|\mathbf{\Sigma}^{-1/2}(\mathbf{x}-\boldsymbol{\mu})\|_2}$ is the CDF of the radial component of the whitened random vector, and $F_\chi^{-1}$ is the inverse CDF of the $\chi(d)$ distribution. We denote the pushforward measure by $T_{\text{VCReg}\#}P_{\mathbf{X}}$ and $T_{\text{Radial-VCReg}\#}P_{\mathbf{X}}$. Let $\mathcal{F}_{\text{VCReg}} = \{P_{\mathbf{X}} : T_{\text{VCReg}\#}P_{\mathbf{X}} = \mathcal{N}(\mathbf{0}, \mathbf{I})\}$ and $\mathcal{F}_{\text{Radial-VCReg}} = \{P_{\mathbf{X}} : T_{\text{Radial-VCReg}\#}P_{\mathbf{X}} = \mathcal{N}(\mathbf{0}, \mathbf{I})\}$ be sets of distributions that can be Gaussianized by the VCReg map and the Radial-VCReg map respectively. Then $\mathcal{F}_{\text{VCReg}} \subsetneq \mathcal{F}_{\text{Radial-VCReg}}$.*

## 3 Synthetic Data Experiments

To test whether Radial-VCReg encourages Gaussianity, we construct the X-distribution in 2D Euclidean space as shown in Figure 1a. Although it has identity covariance, minimizing variance and covariance losses, the distribution is not elliptically symmetric and exhibits higher-order dependencies.

We apply gradient descent over samples from the X-distribution by differentiating the Radial-VCReg loss with respect to the sampled points. In Figure 1b, we show the final samples after 200000 training steps. The resulting points spread spherically and resemble standard normal samples (Figure 1b). We further measure the Wasserstein distance between optimized samples from a mixture $\alpha \mathbf{X} + (1 - \alpha)\mathcal{N}(\mathbf{0}, \mathbf{I})$ and $\mathcal{N}(\mathbf{0}, \mathbf{I})$. As $\alpha$ increases, Radial-VCReg consistently produces samples closer to Gaussian than standard VCReg (Figure 1c). Thus, even though the X-distribution is not elliptically symmetric, the added radial Gaussianization term can push the samples closer to a Gaussian distribution. We also provide additional details and experiments in Appendix 8.

Table 1: **CIFAR-100 Results (Linear Probes).** The table reports the mean $\pm$ standard deviation for Top-1 and Top-5 accuracies, with the two metrics separated by a forward slash (*/*). All results were averaged over multiple random seeds. Hyperparameter details are provided in Appendix 10.1.

| Architecture | Method | Projector Dimension ($d$) | |
| --- | --- | --- | --- |
| | | 512 | 2048 |
| ResNet18 | Radial-VICReg | $\mathbf{65.99 \pm 0.08} / \mathbf{89.28 \pm 0.21}$ | $\mathbf{68.25 \pm 0.41} / 90.61 \pm 0.23$ |
| | VICReg | $64.23 \pm 0.10 / 88.32 \pm 0.10$ | $67.99 \pm 0.27 / \mathbf{90.78 \pm 0.05}$ |
| ViT | Radial-VICReg | $\mathbf{61.33 \pm 0.29} / \mathbf{87.36 \pm 0.28}$ | $\mathbf{62.91 \pm 0.20} / \mathbf{88.11 \pm 0.37}$ |
| | VICReg | $60.30 \pm 0.21 / 86.68 \pm 0.05$ | $62.28 \pm 0.33 / 87.97 \pm 0.31$ |

Table 2: **ImageNet-10 Results (Linear Probes).** The table reports the mean ± standard deviation for Top-1 and Top-5 accuracies, which are separated by a forward slash (*/*). All results were averaged over multiple random seeds. Hyperparameter details can be found in Appendix 10.2.

| Projector Dimension | 512 | 2048 | 8192 |
| --- | --- | --- | --- |
| Radial-VICReg | $\mathbf{94.73 \pm 0.58}/\mathbf{99.27 \pm 0.12}$ | $93.93 \pm 0.31/99.07 \pm 0.12$ | $93.33 \pm 0.70/\mathbf{99.47 \pm 0.23}$ |
| VICReg | $93.20 \pm 0.69/99.07 \pm 0.31$ | $93.53 \pm 0.23/\mathbf{99.47 \pm 0.31}$ | $\mathbf{93.33 \pm 1.55}/99.20 \pm 0.00$ |

Table 3: **CIFAR-100 MLP Probe Results.** The table reports the mean and standard deviation for Top-1 and Top-5 accuracies, which are separated by a forward slash (*/*). All results were averaged over multiple random seeds, and the experimental settings are identical to those in Table 1.

| Projector Dimension | 512 | 2048 |
| --- | --- | --- |
| Radial-VICReg | $\mathbf{64.11 \pm 0.14} / \mathbf{86.88 \pm 0.14}$ | $\mathbf{66.33 \pm 0.33} / 88.05 \pm 0.10$ |
| VICReg | $62.30 \pm 0.34 / 85.58 \pm 0.14$ | $65.81 \pm 0.16 / \mathbf{88.09 \pm 0.42}$ |

## 4   Empirical Results

To evaluate Radial-VICReg, we pretrain networks with 512-dimensional outputs and an MLP projector on CIFAR-100 and ImageNet-10, reporting results in Table 1, 2. Radial-VICReg consistently outperforms VICReg by about $1.5\%$ on both datasets for smaller projector dimensions like 512, with gains holding across ResNet18 and ViT backbones. The improvements remain stable under MLP probing (Table 3), suggesting that radial Gaussianization enhances representations rather than exploiting linear probes. Figures 2a, 2b, and 2c show that the added radial term shifts radius distributions toward the Chi distribution, while Figure 2d illustrates that closer alignment with Chi correlates with higher accuracy. We also observe improvements on CelebA for multi-label attribute prediction (Appendix 12); further experimental details are in Appendix 10.

## 5   Conclusion

We introduced Radial-VCReg, a self-supervised method that augments VCReg with a radial Gaussianization loss to align feature norms with a Chi distribution. This extension pushes a broader class of distributions toward Gaussianity than VCReg alone, as shown theoretically and on synthetic data. Experiments on real-world image datasets confirm that the radial term consistently improves performance. While not sufficient for perfect Gaussianity, it highlights the value of higher-order constraints in learning more diverse and informative representations.

## Acknowledgments and Disclosure of Funding

We thank Alfredo Canziani and Ying Wang for helpful discussions and anonymous reviewers for helpful feedback. This work was supported in part by AFOSR under grant FA95502310139, NSF Award 1922658, and Kevin Buehler's gift. This work was also supported through the NYU IT High Performance Computing resources, services, and staff expertise.

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

# 6   Additional Background

In this section, we review key concepts related to information maximization in self-supervised learning.

**Mutual Information**   Self-supervised learning can be viewed as maximizing the mutual information $I(Z; Z')$ between different views $Z$ and $Z'$ of the same input [Bachman et al., 2019]. By definition, $I(Z; Z') = H(Z) + H(Z') - H(Z, Z')$ where $H$ is the entropy function. During training, we would like to minimize the joint entropy $H(Z, Z')$ and maximize the marginal entropies $H(Z)$ and $H(Z')$. In general, it's difficult to directly maximize the marginal entropy due to the curse of dimensionality.

**Maximum Entropy Distribution**   Even if it's hard to maximize entropy in general, some distributions are maximum entropy by default. Given a fixed mean and variance, the Gaussian distribution is the maximum entropy distribution when compared to all other distributions with support over $[-\infty, \infty]$ [Cover and Thomas, 1991]. This fact also extends to high-dimensional cases. In the context of representation learning, maximizing the entropy of the output feature distribution is crucial to preventing representational collapse, where the model learns to map all inputs to a single, trivial point.

**Elliptically Symmetric Density (ESD)**   Given a random vector $\mathbf{x}$ in $d$ dimension with a zero mean, we say that its density $p_X$ is elliptically symmetric if it has the following form:

$$p_X(\mathbf{x}) = c \cdot f\left( -\frac{1}{2} \mathbf{x}^\top \boldsymbol{\Sigma}^{-1} \mathbf{x} \right) \tag{11}$$

where $c$ is the normalization constant, $\boldsymbol{\Sigma}$ is a positive definite matrix, and $f(\cdot) \geq 0$ and $\int_0^\infty f(-r^2/2) r^{d-1} \mathrm{d}r < \infty$ [Lyu and Simoncelli, 2008]. When $\boldsymbol{\Sigma}$ is the covariance matrix and is a scalar multiple of the identity matrix (i.e., $\boldsymbol{\Sigma} = \sigma^2 \mathbf{I}$), the density function is said to be spherically symmetric. A key property of ESDs is that they can always be transformed into a spherically symmetric density by applying whitening (i.e., making the covariance matrix the identity).

In practice, it's difficult to Gaussianize high-dimensional output features without making assumptions. In the following lemma, we provide a sufficient condition for a Gaussian density that relates to the family of elliptically symmetric densities.

**Lemma 2.** *If $\mathbf{x}$ is a random vector in $d$ dimensions with a spherically symmetric density and the random variable $\|\mathbf{x}\|_2$ follows the Chi distribution $\chi(d)$ with $d$ degrees of freedom, then the density function $p(\mathbf{x}) = \mathcal{N}(\mathbf{0}, \mathbf{I}_d)$.*

*Proof.* From Theorem 4.2 in Fourdrinier et al. [2018], we know that the density function for spherically symmetric density only depends on the norm, i.e. $p(\mathbf{x}) = g(\|\mathbf{x}\|_2)$. Let $r = \|\mathbf{x}\|_2$ be the radius. Our goal is to show that $p(\mathbf{x}) = g(r) = \mathcal{N}(\mathbf{0}, \mathbf{I}_d)$.

It's well known that the infinitesimal volume element $d\mathbf{x}$ in spherical coordinate is given by $d\mathbf{x} = r^{d-1} dr d\Omega_d$ where $\Omega_d$ is the surface measure of a unit sphere $\mathbb{S}^d$. It's shown in Fourdrinier et al. [2018] that the surface measure of a unit sphere is

$$\Omega_d(\mathbb{S}^d) = \int_{\mathbb{S}^d} d\Omega_d = \frac{2\pi^{d/2}}{\Gamma(d/2)} \tag{12}$$

Thus the probability distribution can be computed with this new measure

$$P(\mathbf{x} \in B) = \int_B p(\mathbf{x}) d\mathbf{x} \tag{13}$$

$$= \int_0^\infty \int_{\mathbb{S}^d} p(\mathbf{x}) r^{d-1} dr d\Omega_d \tag{14}$$

$$= \int_0^\infty \int_{\mathbb{S}^d} g(r) r^{d-1} dr d\Omega_d \tag{15}$$

$$= \int_0^\infty g(r) r^{d-1} \left( \int_{\mathbb{S}^d} d\Omega_d \right) dr \tag{16}$$

$$= \int_0^\infty \frac{2\pi^{d/2}}{\Gamma(d/2)} g(r) r^{d-1} dr \tag{17}$$

$$\tag{18}$$

Since we marginalize out the angular components, we can define the density for the radial component $r$ to be

$$p_\chi(r) = \frac{2\pi^{d/2}}{\Gamma(d/2)} g(r) r^{d-1} \tag{19}$$

However, we are also constraining $r$ to follow a Chi distribution $r \sim \chi(d)$ with $d$ degree of freedom. This gives us another expression for the radial marginal

$$p_\chi(r) = \frac{r^{d-1}}{2^{\frac{d}{2}-1}\Gamma(\frac{d}{2})} \exp(-\frac{r^2}{2}) \tag{20}$$

We can combine these two expressions to compute $g(r)$ as follows

$$g(r) = \frac{p_\chi(r)\Gamma(d/2)}{2\pi^{d/2}r^{d-1}} \tag{21}$$

$$= \frac{\frac{r^{d-1}}{2^{\frac{d}{2}-1}\Gamma(\frac{d}{2})} \exp(-\frac{r^2}{2})\Gamma(d/2)}{2\pi^{d/2}r^{d-1}} \tag{22}$$

$$= \frac{1}{(2\pi)^{\frac{d}{2}}} \exp(-\frac{r^2}{2}) \tag{23}$$

$$= \frac{1}{(2\pi)^{\frac{d}{2}}} \exp(-\frac{\|\mathbf{x}\|^2}{2}) \tag{24}$$

$$= \mathcal{N}(\mathbf{x}; \mathbf{0}, \mathbf{I}) \tag{25}$$

Thus we have shown that any random vector with spherically symmetric density and Chi-distributed radius with $d$ degree of freedom has to be the standard multivariate normal distribution $\mathcal{N}(\mathbf{0}, \mathbf{I}_d)$. □

Lemma 2 shows that we can transform any distribution from the ESD family into a standard Gaussian by ensuring two conditions are met: isotropic covariance (achieved through whitening) and a Chi-distributed radius. While real-world feature distributions are not guaranteed to be elliptically symmetric, there are cases where this transformation remains useful. We argue that imposing these two conditions serves as a necessary step towards optimizing for Gaussian features, which inherently maximize information content.

## 7 Proofs of Proposition 1

*Proof.* We would like to prove the following equivalent conditions first.

- 1) $T_{\text{VCReg}\#}P_{\mathbf{X}} = \mathcal{N}(\mathbf{0}, \mathbf{I}) \iff P_{\mathbf{X}}$ is Gaussian, i.e., $P_{\mathbf{X}} = \mathcal{N}(\boldsymbol{\mu}, \boldsymbol{\Sigma})$.
- 2) $T_{\text{Radial-VCReg}\#}P_{\mathbf{X}} = \mathcal{N}(\mathbf{0}, \mathbf{I}) \iff P_{\mathbf{X}}$ is elliptically symmetric.

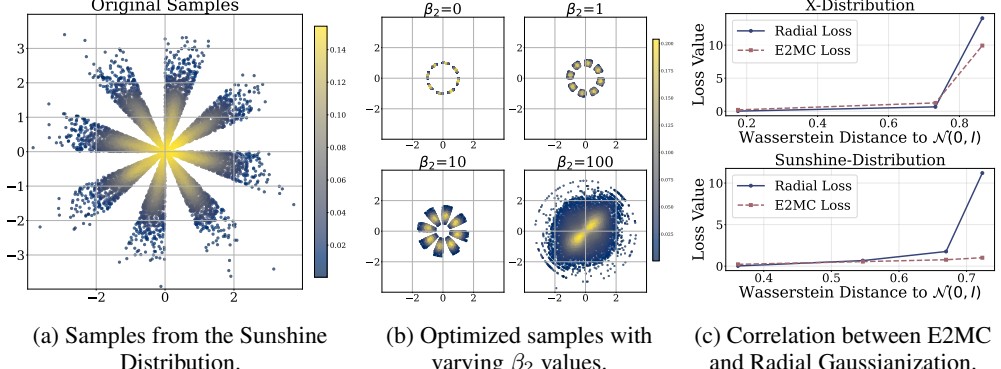

(a) Samples from the Sunshine
Distribution.

(b) Optimized samples with
varying $\beta_2$ values.

(c) Correlation between E2MC
and Radial Gaussianization.

Figure 3: **There exist distributions that minimize the Radial-VCReg loss but are not Gaussian** (a) The sunshine distribution is built by first generating points from a 2D isotropic Gaussian distribution. These points are then converted to polar coordinates and sorted into a specified number of pie slices. Finally, every even-numbered slice is rotated clockwise, creating a distinctive pattern of segmented, rotated clusters. (b) As the weighting $\beta_2$ for the entropy term in the radial Gaussianization loss increases, samples are pushed towards the circle of radius $\sqrt{d-1}$. In 2-dimensions, the radius is just 1. (c) For both the X distribution and the Sunshine distribution, we observe a correlation between the E2MC loss and the radial Gaussianization loss. As both losses decreases, the optimized samples are also closer to a standard normal as measured by the Wasserstein distance.

We list the proofs below for claims 1) and 2).

**Claim 1). VCReg.**

($\Rightarrow$). Since $T_{\text{VCReg}}(\mathbf{X}) \sim \mathcal{N}(\mathbf{0}, \mathbf{I})$, we can write the random vector $\mathbf{X}$ via the affine map $\mathbf{X} = \mathbf{\Sigma}^{1/2} T_{\text{VCReg}}(\mathbf{X}) + \boldsymbol{\mu} \sim \mathcal{N}(\boldsymbol{\mu}, \mathbf{\Sigma})$. Thus $P_{\mathbf{X}} = \mathcal{N}(\boldsymbol{\mu}, \mathbf{\Sigma})$.

($\Leftarrow$). We know that $\mathbf{X} \sim \mathcal{N}(\boldsymbol{\mu}, \mathbf{\Sigma})$. Then the random vector $T_{\text{VCReg}}(\mathbf{X}) = \mathbf{\Sigma}^{-1/2}(\mathbf{X} - \boldsymbol{\mu}) \sim \mathcal{N}(\mathbf{0}, \mathbf{I})$. Thus $T_{\text{VCReg}\#} P_{\mathbf{X}} = \mathcal{N}(\mathbf{0}, \mathbf{I})$.

**Claim 2). Radial-VCReg.**

($\Rightarrow$) We're given that $T_{\text{Radial-VCReg}\#} P_{\mathbf{X}} = \mathcal{N}(\mathbf{0}, \mathbf{I})$. Then $\mathbf{Z} = T_{\text{Radial-VCReg}}(\mathbf{X})$ is spherically symmetric. Let $\mathbf{Y} := \mathbf{\Sigma}^{-1/2}(\mathbf{X} - \boldsymbol{\mu}) = r \cdot \mathbf{\Theta}$, where $r = \|\mathbf{Y}\|_2$ is the radius and $\mathbf{\Theta} = \mathbf{Y}/\|\mathbf{Y}\|_2$ is the angle. Note that $T_{\text{Radial-VCReg}}$ preserves angles and only modifies radius. Therefore, the angular component $\mathbf{\Theta}$ must be uniform and independent of $r$, which implies $\mathbf{Y}$ is spherically symmetric. Hence, $\mathbf{X} = \mathbf{\Sigma}^{1/2}\mathbf{Y} + \boldsymbol{\mu}$ is elliptically symmetric.

($\Leftarrow$) Suppose $P_{\mathbf{X}}$ is elliptically symmetric. Then $\mathbf{Y} = \mathbf{\Sigma}^{-1/2}(\mathbf{X} - \boldsymbol{\mu})$ is spherically symmetric. By Lemma 2, we know that $T_{\text{Radial-VCReg}\#} P_{\mathbf{X}} = \mathcal{N}(\mathbf{0}, \mathbf{I})$.

Now given the equivalent conditions, we know that $\mathcal{F}_{\text{VCReg}}$ consists only of Gaussian distributions, whereas $\mathcal{F}_{\text{Radial-VCReg}}$ contains all elliptically symmetric distributions. Since there exist elliptically symmetric distributions that are *not* Gaussian (e.g., uniform on a sphere or isotropic Student-$t$), we have $\mathcal{F}_{\text{VCReg}} \subsetneq \mathcal{F}_{\text{Radial-VCReg}}$. $\qquad\square$

## 8 Synthetic Distributions

### 8.1 Sunshine Distribution

There also exist non-ESD (elliptically symmetric) distributions that already minimize the Radial-VCReg loss but are not Gaussian. In Figure 3a, we plot the sunshine distribution with an identity covariance matrix and chi-distributed radius. The final optimized samples using the Radial-VCReg objective are shown in Figure 3b with varying weights for the radial entropy loss. Across hyperparameters, Radial-VCReg is unable to push samples from the sunshine distribution towards Gaussian.

This illustrates that certain distributions cannot be fully Gaussianized by the Radial-VCReg objective. Nevertheless, the inclusion of the radial Gaussianization term expands the class of feature distributions that move toward Gaussianity compared to standard VCReg.

In Figure 3c, we explore to what extent the radial Gaussianization loss is related to E2MC [Chakraborty et al., 2025]. We take samples from both the X distribution and the Sunshine distribution with Radial-VCReg optimization and log the corresponding E2MC loss. We find that minimizing the radial Gaussianization loss implicitly leads to a lower E2MC loss. The reduction in both losses also bring samples closer to a standard normal as measured by Wasserstein distances. Therefore, both Radial-VCReg and E2MC are effective proposals for reducing higher-order dependencies and achieving more Gaussian-like samples.

## 8.2  Experimental Details

For both the X-distribution and the sunshine distribution, we utilized a dataset of $10,000$ samples for optimization. Training was performed using stochastic gradient descent (SGD) for $200,000$ steps with a linear warm-up and cosine-decay learning rate scheduler.

We performed a hyperparameter sweep over the following values:

- **Mixture Weight** ($\alpha$): $\{0.01, 0.25, 0.5, 0.75, 0.99\}$
- **Learning Rate**: $\{5 \times 10^{-1}, 5 \times 10^{-2}, 5 \times 10^{-3}, 5 \times 10^{-4}, 5 \times 10^{-5}\}$
- **Radial Gaussianization Parameters** ($\beta_1, \beta_2$): $\{0, 0.1, 1, 10, 100\}$
- **VCReg Parameters** ($\lambda_2, \lambda_3$): $\{1, 10, 25\}$

# 9  Wasserstein Distance Formulation of the Radial Gaussianization Loss

## 9.1  Approximating the Radial Chi Distribution: KL vs. Wasserstein

Our radial objective is one–dimensional: given features $\mathbf{z} \in \mathbb{R}^{d_{\text{out}}}$ with radii $r = \|\mathbf{z}\|_2$, we seek to match the empirical radius distribution $p_\theta^r$ to the Chi distribution with $d_{\text{out}}$ degrees of freedom, denoted $\chi(d_{\text{out}})$. Two natural divergences for this one-dimensional matching are (i) a *KL-based* loss, introduced in the main text, and (ii) a *Wasserstein-1* loss, which we detail here.

**Wasserstein-1 (quantile) radial loss.**  For one-dimensional distributions, the Wasserstein distance is characterized by Vallender [1974]:

$$W_1(p_\theta^r, p_\chi) = \int_{\mathbb{R}} \left| F_\theta^r(t) - F_\chi(t) \right| dt = \int_0^1 \left| (F_\theta^r)^{-1}(u) - (F_\chi)^{-1}(u) \right| du, \qquad (26)$$

where $F$ denotes the cumulative distribution function. We use a simple, low-variance empirical estimator: given $K$ radii samples $\{r_i\}_{i=1}^K$ from the mini-batch and $K$ i.i.d. samples $\{u_i\}_{i=1}^K$ from $\chi(d_{\text{out}})$, we sort both sets and compute

$$\widehat{W}_1 = \frac{1}{K} \sum_{i=1}^K \left| r_{(i)} - u_{(i)} \right|, \qquad \text{with } r_{(1)} \leq \cdots \leq r_{(K)}, \ u_{(1)} \leq \cdots \leq u_{(K)}. \qquad (27)$$

For two augmented views $\mathbf{Z}, \mathbf{Z}'$, we sum their losses:

$$\mathcal{L}_{\text{W1}}(\mathbf{Z}, \mathbf{Z}') = \widehat{W}_1(\{\|\mathbf{z}_i\|_2\}, \chi(d_{\text{out}})) + \widehat{W}_1(\{\|\mathbf{z}_i'\|_2\}, \chi(d_{\text{out}})).$$

We weight the radial Wasserstein term by a scalar $\gamma \geq 0$:

$$\mathcal{L}_{\text{total}}(\mathbf{Z}, \mathbf{Z}') = \underbrace{\lambda_1 s(\mathbf{Z}, \mathbf{Z}') + \lambda_2 [v(\mathbf{Z}) + v(\mathbf{Z}')] + \lambda_3 [c(\mathbf{Z}) + c(\mathbf{Z}')]}_{\mathcal{L}_{\text{VICReg}}(\mathbf{Z}, \mathbf{Z}')} + \gamma \mathcal{L}_{\text{W1}}(\mathbf{Z}, \mathbf{Z}'). \qquad (28)$$

The estimator in eq. (27) is differentiable almost everywhere (via the sort's subgradient routing). Unlike the KL-based loss, however, the Wasserstein-1 estimator depends on the batch size: larger $K$ reduces quantile noise and yields sharper shape matching.

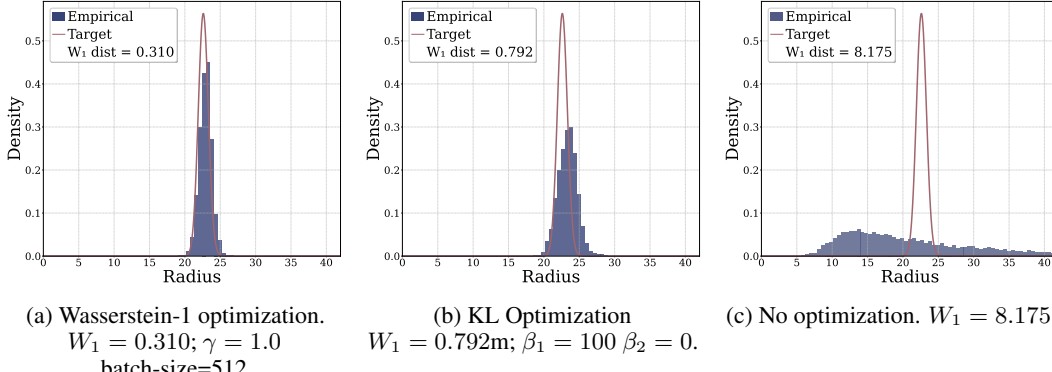

(a) Wasserstein-1 optimization. $W_1 = 0.310; \gamma = 1.0$ batch-size=512

(b) KL Optimization $W_1 = 0.792\text{m}; \beta_1 = 100 \; \beta_2 = 0.$

(c) No optimization. $W_1 = 8.175.$

Figure 4: **Radial Gaussianization aligns radii distributions with the $\chi$ distribution.** Comparison of (a) direct Wasserstein-1 optimization, (b) Radial-VICReg optimization, and (c) no optimization. Both Wasserstein-1 and Radial-VICReg push the empirical radii distribution closer to the target $\chi$ distribution, with Radial-VICReg achieving a substantial improvement over the unoptimized baseline.

**Empirical comparison.** In practice, we find that both KL and Wasserstein objectives optimize essentially the *same* radial constraint. To illustrate this, we compare three cases: (a) optimization directly minimizing the Wasserstein-1 distance, (b) Radial-VICReg optimization using the KL-based radial Gaussianization loss, and (c) no optimization. The results are shown in Figure 4: Wasserstein-1 minimization achieves a distance of $0.310$ to the $\chi$ distribution, KL optimization achieves $0.792$, while the unoptimized baseline achieves a distance of $8.175$.

## 10 Experimental Details

For hyperparameter sweeps, we varied the base learning rate $\{0.3, 0.03\}$, the cross-entropy (CE/rlw) weight $\{0, 1, 10, 100\}$, and the entropy (rlew) weight $\{0, 0.1, 0.3, 0.5, 0.75, 1.0\}$, each across three random seeds.

**CIFAR-100 (ResNet-18).** For all experiments on CIFAR-100 with ResNet-18, we trained the `radialvicreg` method using a three-layer MLP projector with dimensionality varying across settings. We applied standard image augmentations: random resized crops (scale range 0.2–1.0), color jitter (brightness 0.4, contrast 0.4, saturation 0.2, hue 0.1, applied with probability 0.8), random grayscale (probability 0.2), horizontal flips (probability 0.5), and solarization (probability 0.1). Gaussian blur and histogram equalization were disabled for CIFAR-100. Images were resized to $32 \times 32$, with two crops per image. Optimization used LARS with batch size 256, base learning rate (either 0.3 or 0.03 depending on sweep setting), classifier-head learning rate 0.1, weight decay $10^{-4}$, learning-rate clipping, $\eta = 0.02$, and bias/normalization parameters excluded from weight decay. We used a warmup cosine schedule for learning-rate annealing. Training ran for 400 epochs with mixed precision (`fp16`) and distributed data parallelism (`ddp`) across GPUs. The invariance, variance, and covariance loss weights were fixed at 25.0, 25.0, and 1.0, respectively.

**CIFAR-100 (ViT-Tiny/16).** We also trained a vision transformer variant using the `ViT-Tiny/16` architecture from `timm`, consisting of 12 transformer encoder layers with an embedding dimension of 192 and 3 attention heads per layer. For CIFAR-100, we adapted the patch size from 16 to 4 to accommodate $32 \times 32$ images, yielding $8 \times 8$ patches. The projector was configured with hidden and output dimensions of 2048. Optimization employed AdamW with a base learning rate of $5 \times 10^{-4}$ (and $5 \times 10^{-3}$ for the classifier head), batch size 256, weight decay $10^{-4}$, and a warmup cosine learning rate schedule. Training details otherwise matched the ResNet-18 CIFAR-100 setup.

**ImageNet-10 (ResNet-18).** For ImageNet-10, we used a ResNet-18 backbone with a three-layer MLP projector. Images were cropped to $224 \times 224$ and augmented with the same transformations as above, except that Gaussian blur (probability 0.5) was enabled. Optimization followed the CIFAR-100 ResNet-18 settings, except with batch size 128. Training was conducted for 400 epochs with synchronized batch normalization, mixed precision, and two GPUs.

Table 4: **CelebA Multi-Label Classification.** We compare standard VICReg with Radial-VICReg by ablating the cross entropy and entropy terms in the KL divergence for the Chi-distribution. Radial CE stands for only using the cross entropy term, and Radial ENT represents using the entropy term alone. Radial KL uses both with non-zero hyperparameter values for $\beta_1$ and $\beta_2$.

| | Encoder Linear Probe | | Projector Linear Probe | |
|---|---|---|---|---|
| Projector Dimension | 512 | 2048 | 512 | 2048 |
| VICReg | $62.29 \pm 0.49$ | $65.93 \pm 0.35$ | $62.88 \pm 0.58$ | $\mathbf{67.50 \pm 0.39}$ |
| VICReg + Radial CE | $\mathbf{63.37 \pm 0.89}$ | $\mathbf{66.07 \pm 0.27}$ | $\mathbf{64.33 \pm 0.70}$ | $67.48 \pm 0.50$ |
| VICReg + Radial ENT | $50.51 \pm 0.41$ | $54.95 \pm 1.16$ | $50.04 \pm 0.07$ | $55.97 \pm 1.56$ |
| VICReg + Radial KL | $62.40 \pm 0.45$ | $66.00 \pm 0.31$ | $62.76 \pm 0.47$ | $66.54 \pm 0.52$ |

All experiments (on synthetic and image datasets) were run on NVIDIA V100, RTX8000, or A100 GPUs.

## 10.1 Table 1 Details

*ResNet-18.* Best Radial-VICReg hyperparameters on CIFAR-100:

- $d = 2048$: $\beta_1 = 1.0$, $\beta_2 = 0.10$, learning rate $= 0.3$.
- $d = 512$: $\beta_1 = 100$, $\beta_2 = 0.0$, learning rate $= 0.3$.

For VICReg, the best learning rates were 0.03 at $d = 512$ and 0.3 at $d = 2048$. These values were obtained from the sweep described above.

*ViT-Tiny/16.* Best Radial-VICReg hyperparameters:

- $d = 512$: $\beta_1 = 100.0$, $\beta_2 = 0.0$.
- $d = 2048$: $\beta_1 = 1.0$, $\beta_2 = 0.10$.

For VICReg, both $\beta_1$ and $\beta_2$ are set to 0.

## 10.2 Table 2 Details

*ResNet-18.* Best Radial-VICReg hyperparameters on ImageNet-10:

- $d = 512$: $\beta_1 = 100$, $\beta_2 = 0$.
- $d = 2048$: $\beta_1 = 1$, $\beta_2 = 0.5$.
- $d = 8192$: $\beta_1 = 0$, $\beta_2 = 0.1$.

# 11 Additional Results for CIFAR-100

In Figure 5, We show the sensitivity to hyperparameters for the Radial-VICReg objective.

# 12 Additional Results for CelebA

In Table 4, we show the averaged multi-label attributes prediction performances over the CelebFaces Attributes Dataset (CelebA) [Liu et al., 2015] for Radial-VICReg and VICReg. The hyperparameter settings are inherited from the CIFAR-100 experiments. In addition, we sweep the base learning rate in $\{(0.3, 0.03, 0.003)\}$ with linear probe learning rate $\{0.1, 0.01, 0.001\}$. For CelebA, we apply standard data augmentations commonly used in self-supervised learning. Each image is randomly resized and cropped to $128 \times 128$ pixels with scale sampled uniformly from $[0.5, 1.0]$, producing two views per image. We further apply color jittering (brightness/contrast $\pm 0.4$, saturation $\pm 0.2$, hue $\pm 0.1$) with probability 0.8, random grayscale conversion with probability 0.2, Gaussian blur with probability 0.5, and horizontal flipping with probability 0.5. Solarization and histogram equalization

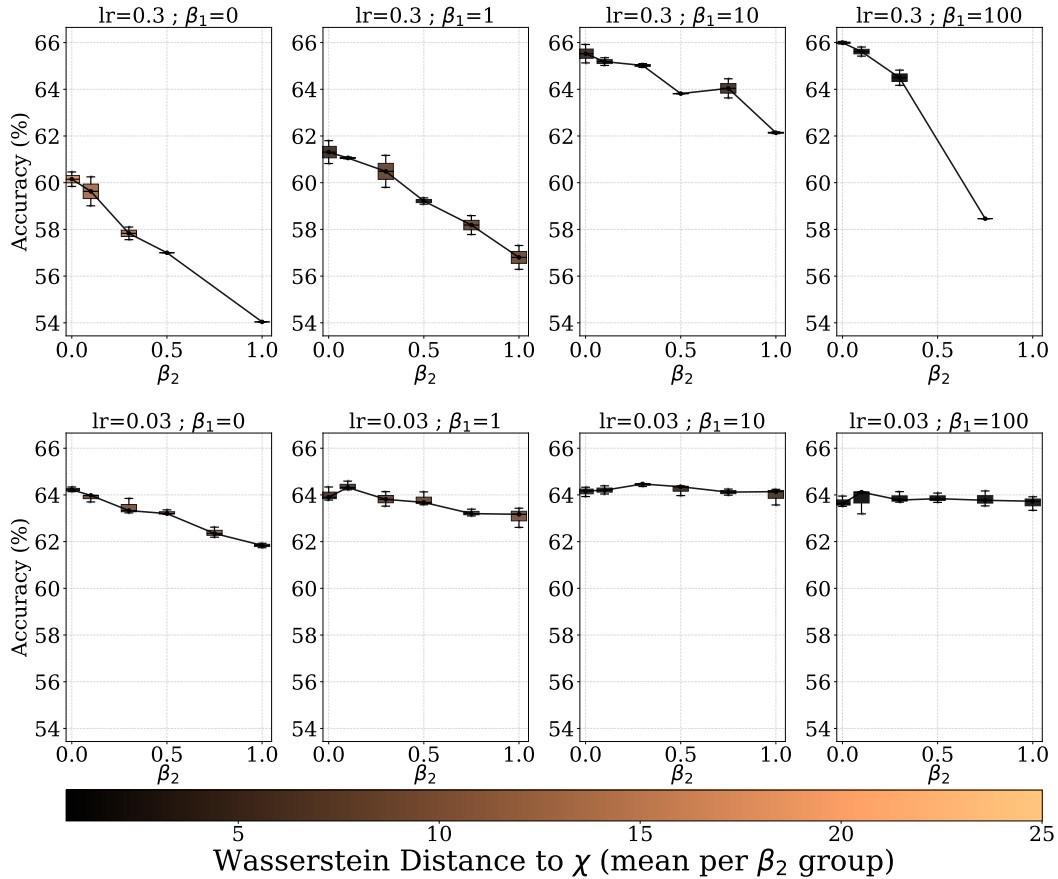

Figure 5: **The optimal performance of Radial-VICReg can be obtained with $\beta_1 \neq \beta_2$, even if $\beta_1 = \beta_2$ gives theoretically consistent estimator of the underlying KL divergence.** We observe that sometimes it's better to have $\beta_1 > \beta_2$ for optimal performance in downstream tasks.

are disabled, as such transformations might distort facial structures and yield unnatural artifacts on human faces.

On average, we observe the most improvements from optimizing the cross entropy term alone in the radial Gaussianization loss. We notice that optimizing the entropy term alone actually leads to a performance degradation. This is expected since maximizing the entropy alone leads to unconstrained variance in the feature norm.

