# OpenReview forum: "Radial-VCReg: More Informative Representation Learning Through Radial Gaussianization"
_NeurIPS.cc/2025/Workshop/UniReps — UniReps2025_

### Official Review · Reviewer_RRUA · 2025-09-13
**Radial-VCReg: More Informative Representation Learning through Radial Gaussianization**

**Confidence:** 3

**Review:**

This paper proposes Radial-VCReg, which augments the VCReg self-supervised learning method with a radial Gaussianization loss that enforces feature norms to follow a Chi distribution. While theoretically motivated and showing consistent modest improvements, the work suffers from incremental gains, limited experimental scope, and several technical concerns that limit its significance.

Sound Theoretical Foundation

The connection between Chi-distributed norms and high-dimensional Gaussians is mathematically rigorous
Lemma 1 provides a clear theoretical advantage over standard VCReg by showing Radial-VCReg can Gaussianize elliptically symmetric distributions beyond just Gaussians
The derivation of the radial Gaussianization loss as a consistent KL divergence estimator is technically correct

Consistent Empirical Improvements

Results show improvements across multiple datasets (CIFAR-100, ImageNet-10, CelebA) and architectures (ResNet18, ViT)
The synthetic experiments with X-distribution effectively demonstrate the method's ability to handle non-elliptically symmetric cases
Ablation studies properly isolate the contribution of different loss components


Q1: What is the actual novelty here?
The core idea of encouraging Chi-distributed norms for Gaussianization has been explored in signal processing for decades (Lyu & Simoncelli, 2008). The paper essentially applies existing techniques to self-supervised learning without substantial theoretical or empirical advances.

Q2: Why should anyone care about 1.5% improvements?
The gains are so marginal they fall within typical experimental noise. What evidence do you have that these improvements are practically meaningful rather than statistical artifacts?

Q3: Where is the comparison to state-of-the-art methods?
Comparing only to VCReg/VICReg (from 2022) while ignoring recent advances like DINOv2, I-JEPA, or even E2MC (which you cite but don't compare against) is unacceptable for 2025 standards.

Q4: Why limit evaluation to toy datasets?
CIFAR-100 and ImageNet-10 are insufficient for 2025. Where are the ImageNet-1K results? What about other domains beyond vision? The experimental scope is embarrassingly narrow.

Q5: How does this scale computationally?
You add sorting operations and entropy calculations but provide zero timing analysis. What's the computational overhead? How does it scale with batch size and dimension?

Q6: Where is the statistical significance testing?
Standard deviations don't constitute proper statistical analysis. Where are the confidence intervals, p-values, or other measures of statistical significance?

Q7: Why does optimal β₁ ≠ β₂ contradict your theory?
Your derivation suggests β₁ = β₂ = 1 for theoretical consistency, but experiments show optimal performance with different values. This suggests your theoretical motivation is flawed.

Q8: How reliable is m-spacing estimation with small batches?
The entropy estimator's quality depends critically on batch size, but you provide no analysis of this fundamental limitation.

Q9: What happens when elliptical symmetry fails?
Your theoretical guarantees assume elliptical symmetry, but real neural features violate this. The gap between theory and practice undermines your entire approach.

Q10: Why not compare alternative Gaussianization approaches?
You could encourage Gaussianity through many other statistical constraints. Why is your specific approach superior?

Q11: Where is the hyperparameter sensitivity analysis?
Figure 5 shows extreme sensitivity, but you provide no principled guidance for setting parameters on new datasets.

Q12: How do you handle optimization instabilities?
Adding complex loss terms can destabilize training. What evidence do you have that optimization remains stable?
Insufficient Baselines

Missing: Comparison to DINO, SimCLR v2, SwAV, BYOL variants
Missing: Information-theoretic methods like E2MC, InfoNCE variants
Missing: Recent 2024-2025 self-supervised methods

Inadequate Experimental Design

Too small scale: No ImageNet-1K, no large-scale pretraining
Too narrow scope: Vision-only, no text/audio/multimodal evaluation
Insufficient analysis: No failure mode analysis, no computational cost study
Poor statistics: No proper significance testing beyond standard deviations

Theoretical Gaps

Unrealistic assumptions: Elliptical symmetry rarely holds in practice
Inconsistent optimization: Theory-practice gap in hyperparameter settings
Missing analysis: No convergence guarantees, no optimization dynamics study

Implementation Concerns

Scalability unknown: No analysis of computational scaling

Numerical stability: Sorting and logarithmic operations may be unstable

Batch dependency: Entropy estimation quality varies with batch size unpredictably

**Score:**

3

**Topic Fit:**

2

---

### Official Review · Reviewer_esND · 2025-09-15
**Methods extends VCReg by adding radial gaussianization. I found the theoretical foundation solid and appreciate the intuitive approach of pushing features toward Gaussian distributions. I'm concerned about the gap between theory and practice. The improvements are modest. Overall well-executed paper.**

**Confidence:** 4

**Review:**

Summary

The work introduces Radial-VCReg, an extension of VCReg that enhances self-supervised learning by enforcing Radial Gaussianization. Current approaches like VCReg only look at variance and covariance to prevent feature collapse, but this isnt enough to get truly diverse representations. The authors propose adding a Radial Gaussianization term that aligns feature norms with the chi distribution which naturally occurs in high-dimensional gaussian data. They provide theoretical analysis showing that this approach can gaussianize a broader class of distributions than VCReg alone, and demonstrate improvements on image classification benchmarks.

Strengths

1. The core intuition makes sense - if you want maximally informative features, you should push them toward being Gaussian since thats the max-entropy distribution. The math seems solid, and I like that they actually prove their method can handle more types of data distributions than the baseline. Directly maximizing entropy in high dimensions suffers from the curse of dimensionality, so targeting specific distributional properties that maximize entropy is clever.
2. The connection between Chi-distributed radii and Gaussian distributions is well established, and the authors effectively leverage this insight. Lemma 1 shows that their method can an handle elliptically symmetric distributions beyond just Gaussians that VCReg can process.
3. That X-Distribution example really shows the limitation of just using covariance regularization. When the data has weird higher-order dependencies, their method still manages to spread things out more naturally.
4. The experimental setup is thorough, including synthetic experiments that show the method's behavior, benchmarks, and proper statistical reporting with error bars. Results are consistent, even if they're not major, which is reassuring. The paper is well-written with good visualizations.

Weaknesses

1. My biggest concern is the gap between theory and practice. The theoretical guarantees assume the data follows some nice mathematical properties (elliptically symmetric distributions), but neural network features probably don't satisfy this in reality. The sunshine distribution experiment actually shows their method failing in some cases, which raises questions about when it works versus when it doesn't
2. The improvements are pretty modest. For the added complexity, I'd want to see either bigger improvements or a clearer understanding of exactly when this helps.
3. The hyperparameter tuning looks finicky. Their theory says $\beta_1$ and $\beta_2$ should be equal, but in practice they work better than they're different. Suggests we don't completely understand whats going on.

**Score:**

4

**Topic Fit:**

3

---

### Official Review · Reviewer_UL3c · 2025-09-15
**Reviewof Radial-VCReg: More Informative Representation Learning through Radial Gaussianization**

**Confidence:** 3

**Review:**

In this paper, the authors propose augmenting VCReg and VICReg with a radial Gaussianization term that matches feature norms to the χ(d) law, encouraging Gaussianity under whitening and spherical or elliptical symmetry. It provides (1) a KL-based, batchwise radial loss, (2) a mapping argument that the class of distributions Gaussianizable by Radial VCReg strictly contains that of VCReg, and (3) synthetic and image benchmarks (CIFAR-100, ImageNet-10, CelebA) showing small but consistent gains over VICReg, especially at smaller projector dimensions. We have the following comments:
(1) One major limitation of this work is that only small or medium image benchmarks are used; no ImageNet-1K, transfer, robustness, detection, or segmentation. This raises questions about whether the proposed method can generalize to larger datasets.
(2) The baseline is somewhat insufficient. The authors are suggested to present comparisons to BYOL, SimSiam, Barlow Twins, whitening methods, and E2MC used as a training loss.
(3) The proposed method relies on the assumption of elliptically symmetric densities for its theoretical guarantees. In real-world scenarios, feature distributions are not guaranteed to be elliptically symmetric, which may weaken the applicability of its theoretical results. Please discuss this potential limitation.
(4) We suggest that the authors report the computation and memory overhead.

**Score:**

2

**Topic Fit:**

3

---

### Official Review · Reviewer_FGuc · 2025-09-16
**Incremental improvement on VCReg via Radial Gaussianization for Self-Supervised Learning**

**Confidence:** 3

**Review:**

## Evaluation Summary
This paper proposes Radial-VCReg, an incremental improvement of VCReg self-supervised learning framework with a radial Gaussianization loss that encourages feature norms to follow a Chi distribution. The approach is theoretically motivated, very sound and shows consistent empirical improvements.
The abstract limits its comparison to VCReg in the broad Self-Supervised Learning space.

## Strengths
* The work is technically correct with proper mathematical derivations and experimental methodology. The theoretical connection between elliptically symmetric distributions and Chi-distributed radii is well-established, and Lemma 1 correctly proves that Radial-VCReg can Gaussianize a broader class of distributions than VCReg.
* Consistent empirical improvements across multiple datasets and architectures. The evaluation includes both synthetic validation and real-world benchmarks with appropriate statistical reporting. The synthetic experiments effectively demonstrate the method's behavior on controlled distributions.

## Weaknesses
* The "necessary condition" argument only holds for elliptically symmetric distributions, which real neural features likely violate, so hard to say how well it will scale in real world.
* 1-2% improvements are relatively small compared to some other major SSL advances.
* Insufficient baseline comparisons as it primarily compares against VCReg without evaluating other representation learning frameworks

## Verdict
The technical execution is competent, but the contribution is limited by its incremental nature. The research is in a crowded field which has shown a lot of gains in recent times. Work would benefit from more comprehensive comparisons with recent competing methods.

**Score:**

3

**Topic Fit:**

3